# Field evaluation of drone and AI assisted larval source management in Ghana

**Godfred A. Bokpin[1], Francis A. Adzei[1,2], Samuel Dadzie[3], Masaki Umeda[4], Juhoe Kim[4]\***

**1** University of Ghana Business School, Accra, Ghana, **2** Takemi Program in International Health, Havard School of Public Health, Boston, Massachusetts, United States of America, **3** Noguchi Memorial Institute of Medical Research, University of Ghana, Accra, Ghana, **4** SORA Technology ltd, Nagoya, Japan

\* juhoe.kim@sora-tech.com

## Abstract

### Background

Malaria remains a major public health burden in sub-Saharan Africa. In Ghana, in particular, larval source management (LSM) is increasingly recognized as a complementary vector control strategy. This study evaluates a field-adapted LSM approach that integrates drone-based mapping and artificial intelligence (AI)–driven site prioritization to enhance operational efficiency and reduce resource use.

### Methods

The intervention replaces conventional manual scouting with aerial mapping conducted one day prior to larvicide application. An AI model analyzes geospatial and morphological features of water bodies to identify high-risk larval habitats. Site coordinates are transmitted to field teams via mobile devices for targeted treatment. A comparative field trial was conducted in eight administrative sub-districts within Ghana's Eastern Region. Four sub-districts implemented the drone- and AI-assisted approach, while four served as controls using standard LSM procedures. A mixed-methods evaluation was employed, incorporating quantitative metrics and qualitative field insights.

### Results

Drone-assisted mapping led to more than a threefold increase in the number of identified breeding sites. AI-based targeting reduced larvicide consumption by over 60%. The combined technologies lowered worker requirements by approximately 50%. Despite these reductions, malaria case trends in the intervention sub-districts remained comparable to those in the control sub-districts. The study's limitations include its restriction to the dry season and below-average rainfall, which may have influenced mosquito abundance and transmission.

**Data availability statement:** The data supporting the findings of this study are available in Zenodo at https://doi.org/10.5281/zenodo.18085478.

**Funding:** This study was supported by the Japan International Cooperation Agency (JICA) and the Japan Science and Technology Agency (JST: Grant Number JPMJTT23J4) under the framework of international technical cooperation. Funding was administered through SORA Technology Ltd. Using this funding, SORA Technology Ltd., through its affiliated authors Masaki Umeda (MU) and Juhoe Kim (JK), contributed to the study design, implementation, and manuscript preparation. The funder also provided institutional research support for the affiliations of Godfred A. Bokpin (GAB) and Francis A. Adzei (FAA). The specific roles of these authors are articulated in the 'author contributions' section. JICA and JST had no role in the study design, data collection and analysis, decision to publish, or preparation of the manuscript. There was no additional external funding received for this study.

**Competing interests:** Authors Masaki Umeda (MU) and Juhoe Kim (JK) are employed by SORA Technology Ltd. Through funds administered by SORA Technology Ltd., the affiliations of Godfred A. Bokpin (GAB) and Francis A. Adzei (FAA) received institutional research support. Authors affiliated with SORA Technology contributed to the study design, implementation, and manuscript preparation in their capacity as academic contributors. This does not alter our adherence to PLOS ONE policies on sharing data and materials. All other authors declare no competing interests.

## Conclusions

Drone- and AI-assisted LSM demonstrated substantial resource savings without compromising vector control outcomes. Further longitudinal evaluation across transmission seasons is warranted to assess sustained effectiveness and inform national policy.

## Introduction

Global efforts to control malaria have long relied on integrated vector management strategies, including long-lasting insecticidal nets (LLINs), indoor residual spraying (IRS), and prompt treatment of clinical cases [1–3]. Yet, malaria remains a significant global health threat. In 2023, an estimated 263 million cases and 597000 deaths were reported across 85 countries, with sub-Saharan Africa accounting for approximately 94% of all cases and 95% of deaths [4–6]. The burden disproportionately affects vulnerable populations, particularly children under five and pregnant women, while socioeconomic and geographic inequities further exacerbate barriers to access. Children from the poorest households in sub-Saharan Africa are five times more likely to be infected than those from wealthier households [5].

Despite intensified global action, recent trends show stagnation in malaria control, threatening the achievement of Sustainable Development Goal 3 targets. In response, the WHO's Global Technical Strategy for Malaria 2016–2030 aims to reduce incidence and mortality by 90% by 2030 through three pillars: scaling up responses, increasing investment, and promoting innovation. One such innovation is larval source management (LSM), which targets mosquito breeding sites through larviciding, environmental manipulation, and habitat modification [7,8]. LSM also includes biological control agents such as larvivorous fish. Gambusia affinis has long been recognized as an efficient predator of mosquito larvae in permanent aquatic habitats, and in Ghana, studies have examined the prey selectivity and captive-breeding feasibility of the native lampeye killifish (Aplocheilichthys spilauchen) for potential use in mosquito bio-control [9,10]. However, in large-scale LSM targeting densely populated areas, many breeding sites are temporary pools where fish cannot survive, making wide-scale deployment of fish neither practical nor sustainable.

LSM has long been recognized as an effective vector control strategy, particularly in low-transmission and elimination settings, with documented successes dating back to the early 20th century in several countries [11,12]. However, LSM implementation has remained limited especially in high-burden areas such as sub-Saharan Africa, mainly because of inefficient and labor-intensive waterbody mapping and treatment [8,13,14].

In response, there is growing interest in enhancing LSM performance through geospatial technologies including satellite imagery [15,16], drones [17–21], and AI [16,19,22]. While mosquito species such as *Anopheles* and *Aedes* differ in their ecological preferences, many of their breeding sites are small and temporary, making high-resolution drone imagery more suitable than satellite imagery for detection [13, 14].

Although many existing studies have focused on the technical feasibility of geospatial tools, such as the detection accuracy and classification performance of drone or AI based systems, few have systematically assessed their real-world operational implications. Specifically, little evidence exists on how these technologies influence the actual efficiency of larvicide application, programmatic cost-effectiveness, or reductions in the burden of mosquito-borne diseases such as malaria, when implemented at scale in sub-Saharan Africa. A rare example is the study by Vigodny et al. [23], which evaluated a satellite- and mobile-enabled operational LSM platform in São Tomé and Príncipe. While they report a 74.9% reduction in larval-positive sites and a 52.5% decline in malaria incidence, the comparison was made between areas without any LSM implementation and those where their system-supported LSM was introduced. No direct comparison was made between their system and conventional manual LSM approaches. Moreover, no prior study has evaluated an LSM approach that integrates both drone- and AI-assisted habitat mapping with the actual implementation of larvicide application, as proposed in the present study. While previous studies have attempted automated detection of potential breeding sites from drone imagery using image recognition AI, such approaches remain neither standardized nor consistently reliable for field deployment. Our study therefore focused on advancing beyond simple detection by integrating AI-based risk classification into a three-step pipeline of drone imagery, waterbody identification, and operational implementation. This operational integration of risk classification represents the novelty of our work and provides practical evidence of its value in improving the efficiency of larval source management.

Ghana remains endemic for malaria. While confirmed cases decreased from 5.8 million in 2021 to 5.2 million in 2022, and malaria-related deaths dropped from 2,799 in 2012–151 in 2022, the disease continues to pose a substantial public health challenge [24]. Prevalence has also declined from 27.5% in 2011 to 8.6% in 2022, but children under five and pregnant women remain most at risk [25]. In line with the WHO strategy, Ghana has transitioned from a control-oriented approach to a national malaria elimination agenda. LSM has been introduced in selected districts since 2019, yet uptake has been slow due to high operational costs and logistical constraints. The WHO guidelines emphasize that decisions to implement LSM should consider vector bionomics, larval ecology, epidemiological context, and cost-effectiveness [7].

To explore whether innovative technologies can help overcome operational barriers in LSM, this study was implemented in two ecologically similar sub-districts within the Kwaebibirem District, Eastern Region of Ghana, and evaluated both feasibility and cost benefit of the drone and AI assisted approach compared to conventional manual methods. Unlike highly technical research prototypes, the intervention was designed to be practical and field-operable, relying on local staff trained through brief, structured sessions. The multi-month pilot study encompassed the entire operational cycle, from larvicide application to outcome assessment. By situating the project within a collaboration between academic and public health institutions and emphasizing operational realism, the study offers neutral and policy-relevant insights into the scalability and practical utility of emerging technologies for LSM in resource-constrained settings.

## Methods

This study aimed for evaluating the effectiveness, efficiency, and cost-benefit of drone- and AI-assisted LSM compared to conventional manual methods. To achieve this goal, a comprehensive mixed-methods approach was used. The research was conducted across eight sub-districts in Kwaebibirem District in Ghana's Eastern Region, comprising four intervention sub-districts and four matched control sub-districts. The intervention and control sub-districts were selected based on comparable ecological and demographic characteristics to ensure fair comparison.

To capture both performance metrics and implementation quality, the evaluation design combined quantitative assessments (e.g., waterbody mapping rate, larvicide usage, labor input) with qualitative field observations and process documentation. This integrated framework enabled a multidimensional analysis of how drone-based mapping and AI-assisted risk classification influence LSM operations in real-world settings.

The intervention involved replacing traditional manual scouting with aerial drone mapping, followed by automated classification of high-risk water bodies using an AI model. High-risk coordinates were transmitted to field teams via mobile devices, streamlining their movements and reducing redundant coverage.

The integration of drone technology and AI into LSM builds on a growing body of literature documenting the operational potential of geospatial tools in vector control. For instance, Hardy et al. demonstrated the feasibility of using low-cost drones for mapping malaria vector habitats in Zanzibar [26], noting their ability to capture high-resolution imagery of small, hard-to-reach breeding sites. Hardy et al. further assessed operational larviciding using drones and smartphones, highlighting improvements in coverage efficiency and resource utilization [27]. Similarly, Stanton et al. evaluated drone applications for larval habitat identification in rural settings and found that drone-based methods significantly outperformed ground-based mapping in detecting potential breeding sites [19].

### Ethical approval and permissions

The study protocol was reviewed and approved by the National Malaria Elimination Program (Ghana) under the Ministry of Health of Ghana, which subsequently issued directives to the district health office and its sub-health provision centers. Additional approval and support were obtained from community leaders, such as the Assembly men in the Kwaebibirem district, prior to field implementation. No deviations from the approved study protocol occurred during the research.

All drone flights were conducted in compliance with Ghana Civil Aviation Authority (GCAA) regulations. Mapping covered entire communities, and while incidental images of humans or livestock may have been captured, these were neither analyzed nor used for any purpose; image review and analysis were strictly limited to the identification of surface waterbodies. Feedback from field staff and local stakeholders was obtained voluntarily as part of routine operational activities, without recording personal identifiers, and therefore did not constitute human subjects research requiring separate institutional review board approval.

### Study site

The study was conducted in eight sub-districts within the Kwaebibirem District, located in Ghana's Eastern Region. These sub-districts were divided into two groups: four intervention sub-districts (Abaam, Asuom, Takyiman, and Nkwantanang) and four control sub-districts (Subi, Kade, Pramkese, and Abodom).

In the intervention sub-districts, a drone and AI assisted LSM approach was implemented for the first time. In contrast, the control sub-districts had an established history of conventional manual LSM, regularly implemented since 2019.

Sub-district selection was carried out in close collaboration with the Ghana National Malaria Elimination Program, which played a key role in identifying feasible and priority areas for field implementation. While the allocation of sub-districts to the intervention and control groups was not randomized, efforts were made to ensure comparability between the groups in terms of population size, surface area, and ecological characteristics.

LSM activities were scheduled such that each week, one sub-district from each group was targeted. Over the course of the study, each sub-district received approximately four weeks of LSM operations. In the intervention group, the drone- and AI-assisted operations were conducted during the dry season, from September to December 2024. In the control group, conventional LSM activities began earlier, in July 2024, and continued through December. This difference in timing, particularly the inclusion of the rainy season in the control sub-districts, may have conferred an advantage in terms of larval detection and should be considered when interpreting comparative results.

Because the intervention and control sub-districts are geographically proximate within the same district, they were subject to broadly similar climatic conditions. However, this proximity also presents the possibility of entomological or epidemiological spillover effects between groups. A summary of the key characteristics of each sub-district, including population, surface area, and geographic location, is presented in Table 1.

**Table 1. Information about sub-districts.**

| Control/ Intervention | Sub-district | Official Population | Official Area [km²] | LSM Starting Month in 2024 |
|---|---|---|---|---|
| Intervention | Nkwantanang | 5070 | 0.94 | September |
| Intervention | Abaam | 7100 | 1.03 | September |
| Intervention | Takyiman | 4999 | 1.77 | September |
| Intervention | Asuom | 12308 | 4.01 | September |
| Intervention | Intervention total | 29477 | 7.75 | – |
| Control | Abodom | 5265 | 0.98 | July |
| Control | Subi | 2695 | 1.11 | July |
| Control | Pramkese | 7016 | 1.55 | July |
| Control | Kade | 25736 | 7.14 | July |
| Control | Control total | 40712 | 10.78 | – |

## Overview of proposed drone and AI integrated LSM

**Conventional LSM approach.** In conventional LSM operations, site mapping is performed through direct visual observation. A lead team member moves through the community to identify potential larval habitats, such as visible waterbody sites with notable land slopes. Local residents are also consulted to locate commonly occurring stagnant water sites. Once sites are identified, the lead records geographic coordinates of the suspected habitats. At each site, GPS coordinates are generated using a mobile application, with attention to the nearest physical landmarks to aid in future localization. Moreover, in conventional LSM, spray planning is typically conducted on the same day as larvicide application, often in parallel with fieldwork. As a result, spray operators have limited visibility into the full scope of their tasks prior to deployment, which can hinder operational efficiency and site coverage. Both the conventional and proposed LSM approaches used the same larvicide product, VectoLex WDG [28], to ensure comparability of treatment effectiveness. VectoLex WDG is a biological larvicide containing Bacillus sphaericus 2362 (strain ABTS-1743), designed for residual control of mosquito larvae with a reported efficacy of 3–4 weeks. It is formulated as a water-dispersible granule that can be applied with standard ground or aerial equipment across diverse aquatic habitats. Importantly, VectoLex is highly specific to mosquito larvae and has minimal impact on non-target aquatic organisms such as fish and beneficial invertebrates, thereby supporting ecological safety [29–34]. In Ghana, larvicide selection and application protocols are determined by the National Malaria Elimination Program (NMEP), under which VectoLex has been adopted as the standard product for operational LSM activities.

**Drone-based mapping.** The proposed method replaces traditional manual scouting with drone-assisted aerial mapping. Prior to flight operations, the mapping area was delineated using Google Earth Pro [35] to define its boundaries and surface area, ensuring complete spatial coverage of the designated LSM zone.

To ensure operational safety, all drone flights were conducted under visual line-of-sight conditions. As it was not feasible to map the entire area in a single flight, the study region was subdivided into a virtual grid. Each grid unit was assigned a center point based on latitude and longitude coordinates, which served as operational waypoints for sequential and efficient flight execution. The specific grid size used for this partitioning is described in a later section.

Drone flights were conducted 1–2 days prior to larvicide application to ensure that mapping reflected the most recent environmental conditions. The drones used in this study were equipped with RGB cameras, capturing data exclusively in the visible spectrum; no multispectral or infrared sensors were employed.

The multiple aerial images captured during these flights were processed using image processing software [36] to generate a single high-resolution orthomosaic image. This orthomosaic provided a geospatially consistent and comprehensive map, which served as the visual foundation for subsequent waterbody identification and AI-based analysis.

**Manual extraction and feature recording with AI assistance.** Following data collection, RGB imagery was manually reviewed to identify surface waterbodies through visual inspection. The presence and location of waterbodies were recorded by assigning GPS coordinates to each visually confirmed site. In addition to positional data, key environmental attributes of each waterbody were also manually assessed and recorded to serve as input features for the subsequent AI-based risk classification. These attributes included surface area, visual turbidity, and the presence or absence of vegetation inside and around the waterbody. Image recognition AI was also used to improve the efficiency of these manual processes, but it was not applied widely due to the constrained computational resources. The performance evaluation of the AI model for waterbody detection is beyond the scope of this study and is therefore not discussed in this paper. Image recognition AI was implemented using the MMDetection framework [37], trained on aerial imagery collected in Accra.

**AI-based risk classification and operational deployment.** Each identified waterbody was analyzed using the other AI model trained to estimate larval risk based on selected environmental attributes. The model labelled risks based on these features, categorizing each as either "high" or "low" to reflect the likelihood of mosquito breeding activity. The risk-classified waterbodies were uploaded to a custom mobile application designed to support field operations. Larvicide applications were carried out by on-ground larvicide spraying teams, guided by navigators who were trained to use the application to locate and prioritize accessible high-risk sites. This approach enabled more targeted interventions while reducing unnecessary coverage of low-risk areas, thereby improving resource efficiency.

**Operational benefits of task separation.** An additional operational advantage of the proposed method lies in the separation of mapping and spraying tasks. By conducting drone-based mapping and AI risk analysis prior to the spraying date, field teams were able to receive pre-defined lists of target sites and an overview of the planned route before deployment. This reduced cognitive burden on spray operators during fieldwork and allowed for more structured and time-efficient operations.

Fig 1 provides an overview of the proposed LSM method.

## Drone flight Protocol

A DJI Mavic 3 Multispectral [38] drone was used for aerial mapping. Although the drone supports both RGB and multispectral imaging, only the RGB camera was used due to practical processing constraints. On-site laptops required approximately three hours to generate an orthomosaic image from RGB data for a single sub-district, while processing multispectral data was estimated to take significantly longer. This rendered the use of multispectral imagery impractical under field conditions.

To ensure visual line-of-sight operation, each flight was limited to a 300 m × 300 m area. Prior to actual operations, test flights were conducted at 50 m and 100 m altitudes to evaluate the trade-off between resolution and efficiency. While 50 m flights provided higher resolution, they required about 20 minutes per grid, whereas 100 m flights took only five minutes and still yielded images sufficient to detect surface waterbodies several tens of centimeters in diameter. At 100 m, the ground sampling distance is approximately 2.6 cm/pixel according to manufacturer specifications. Considering this balance between flight time and image quality, together with operational safety concerns such as terrain variations and the risk of collisions with tall structures like transmission towers, 100 m was adopted as the standard flight altitude. Flights were executed using the drone's default automated mapping pattern, with overlap settings of 80% front and 70% side in accordance with software-recommended defaults for orthomosaic generation.

All drone flights were conducted in compliance with Ghanaian aviation regulations, with flight permissions obtained from the Ghana Civil Aviation Authority (GCAA) and coordinated with district authorities. Prior to operations, community leaders introduced by the NMEP informed residents about the planned activities, and during the study period additional announcements were made through public broadcasting systems in each community.

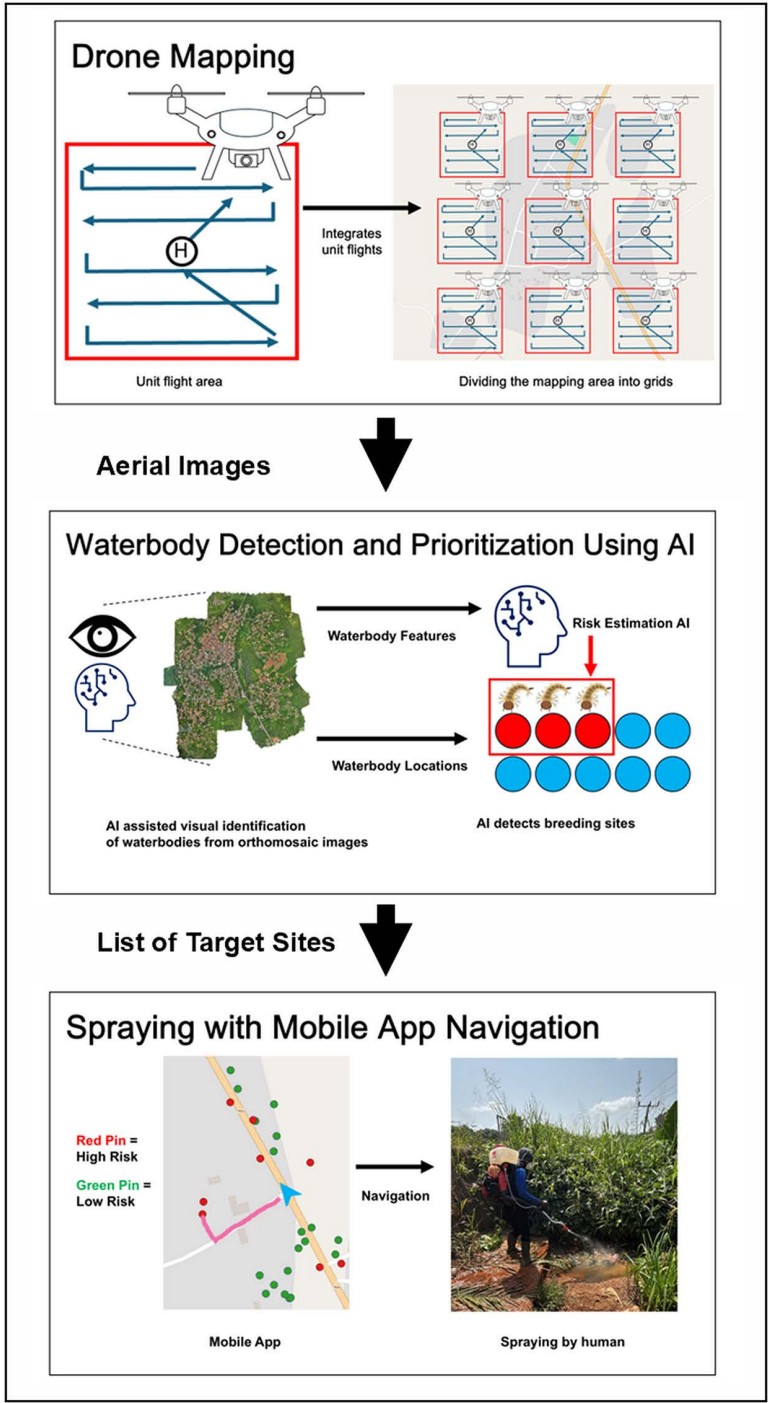

**Fig 1. Overview of proposed LSM method. Background map tiles © OpenStreetMap contributors, licensed under CC BY-SA 2.0. Additional annotations by the authors.**

## Larval risk estimation AI

It was hypothesized that identifying high-risk waterbodies in advance would improve LSM efficiency by minimizing unnecessary larvicide use compared to applying treatments uniformly. While there is no widely accepted model that can explicitly characterize larval habitats, we assumed that a machine learning model trained on paired observations of waterbody features and confirmed larval presence could learn to approximate this relationship.

We employed CatBoost [39], a gradient boosting decision tree algorithm optimized for categorical input. Features were derived exclusively from RGB drone imagery and included six categorical variables: (1) waterbody type, (2) origin (artificial vs. natural), (3) area size class, (4) vegetation inside the waterbody, (5) vegetation around the waterbody, and (6) visual turbidity. These are summarized in Table 2. The waterbody area size class was classified into four groups: Very Small (<0.5 m × 0.5 m), Small (0.5 m × 0.5 m to 1 m × 1 m), Medium (1 m × 1 m to 10 m × 10 m), and Large (>10 m × 10 m).

The contribution of each feature was estimated using feature importance scores derived from the trained CatBoost model. The values reported in Table 2 represent normalized relative feature importance (RFI). Based on RFI, waterbody type and visual turbidity emerged as comparatively influential predictors.

Training data were collected from 4021 waterbodies surveyed between January and August 2024 in Bo District (Sierra Leone) and Greater Accra Region (Ghana). Presence or absence of the mosquito larvae was determined through multiple dipping attempts per waterbody, following the WHO-standard larval sampling protocols [12]. All detections were visually confirmed and photographically documented. While the data collection process followed standardized protocols, certain input features, particularly visual turbidity, were based on field staff judgment due to the impracticality of bringing quantitative instruments into the field. As multiple staff members contributed to data collection, individual biases were likely to average out; nonetheless, subjectivity in turbidity estimation remains a notable limitation of the model.

Larval risk was defined as a binary variable based on the presence of mosquito larvae in waterbodies: sites with at least one larva were classified as high risk (1), while sites without larvae were classified as low risk (0). The AI model was trained as a binary classifier to predict this risk, outputting probabilities between 0 and 1. A fixed decision threshold of 0.5 was applied to assign the final class.

Hyperparameters were tuned empirically with the objective of maximizing recall while minimizing false negatives, since overlooking high-risk habitats was considered to have more severe consequences than overtreatment. Training was initialized with a maximum of 1,000 iterations, but early stopping selected the optimal model at 677 iterations. The maximum tree depth was fixed at 10, which controlled the complexity of each boosting iteration. As no independent external dataset was available at the outset of this study, model performance was evaluated internally using stratified 5-fold cross-validation, in which 80% of the samples were used for training and 20% were held out for testing in each fold while maintaining class balance. This procedure provided a robust internal estimate of generalization performance. The model achieved an average accuracy of 0.70, precision of 0.70, and recall of 0.72.

**Table 2. AI input features.**

| Feature | Variation | Categories | RFI [%] |
|---|---|---|---|
| Waterbody type | 7 | drainage ditches, swampy area, reservoir, puddle made by rain, rice paddy, upland rice field, other agricultural field | 26 |
| Origin | 2 | artificial, natural (seven-types) | 16 |
| Area size class | 4 | four types | 8 |
| Vegetation inside the waterbody, | 2 | present, not present (two-types) | 17 |
| Vegetation around the waterbody | 2 | present, not present (two-types) | 12 |
| Visual turbidity | 3 | clean, turbid, very turbid (three-types) | 21 |

In operational terms, precision influences the degree of overtreatment: higher precision reduces unnecessary treatment of low-risk waterbodies, whereas lower precision increases overtreatment but remains comparable to existing operations, which apply larvicides to all detected waterbodies. By contrast, recall governs the likelihood of missing high-risk habitats, with lower recall carrying greater risk of negative public-health outcomes. For this reason, recall was prioritized in model design, and improvements in precision are pursued only insofar as they do not compromise recall.

## Result collection and evaluation method

To compare the effectiveness and efficiency of the proposed and conventional LSM methods, we collected data across three main categories: (1) operational efficiency, (2) entomological, and (3) epidemiological outcomes.

**1. Operational efficiency data.** This dataset included metrics such as the number of identified and sprayed waterbodies, the quantity of larvicide used, the number of personnel involved in field operations and score of AI estimation. These indicators allowed for comparative assessment of the scale and resource intensity of LSM implementation. Data was recorded in real time by research team members who accompanied both conventional and drone-assisted LSM operations in the field. For the evaluation of operational efficiency, we calculated two indicators separately for the control and intervention groups. Comparisons between groups were conducted using the Mann–Whitney U test. Since the sample size in each group was limited to four, the statistical results should be interpreted as supportive rather than definitive.

The first indicator was Total Person-Days per Unit Area (TPDUA), representing the total number of workdays required per unit area (1 km$^2$) to carry out LSM operations. TPDUA was defined as follows:

$$TPDUA = \frac{1}{S} \sum_{i=1}^{N} W_i D_i$$

(1)

where $S$ denotes the total area of the operation site, $i$ indicates the work phase, $W_i$ the number of workers involved in phase $i$, and $D_i$ the number of workdays for that phase.

The second indicator was the number of waterbodies treated per unit of larvicide used, calculated simply as the number of waterbodies sprayed divided by the number of larvicide packs applied. We did not normalize by the exact mass or concentration of larvicide because such precise measurements were operationally infeasible under field conditions.

**2. Epidemiological data.** To assess the impact of LSM on malaria incidence, outpatient department (OPD) records of confirmed malaria cases were collected from local health facilities across all sub-districts. Data were gathered between September 2024 and mid-January 2025 and provided by the Ghana Health Service through standard reporting channels.

**3. Entomological data.** Entomological surveillance was conducted using CDC light traps, which are widely recognized tools for collecting adult mosquitoes [40]. Captured specimens were morphologically identified to the genus level. The control site, Pramkese, was selected due to its status as an established LSM surveillance zone with prior experience using CDC light traps. The intervention site, Abaam, was chosen for its ecological and spatial similarity to Pramkese, allowing for consistent data collection across sub-districts. Due to resource constraints, it was not feasible to deploy traps across all sub-districts. We therefore based the entomological sampling on subdistricts with prior experience in CDC light-trap surveillance, which ensured reliable implementation and maintained comparability between the intervention and control sites. Data collection was conducted every two days in cooperation with community volunteers, over a four-month period, from 18 September 2024 to 8 January 2025.

Prior to the commencement of surveillance activities, community sensitization campaigns were conducted in both sub-districts to ensure local participation and cooperation. Traps were strategically deployed in areas with reported mosquito nuisance or frequent nighttime outdoor activity. Placement locations were selected between known breeding sites

and residential zones, while avoiding wind-exposed areas, dense vegetation, and competing light sources such as street-lights. Traps were not deployed during rainfall.

Four CDC light traps were deployed at each site, totaling eight traps across both communities. Each site was divided into four quadrants (north, south, east, and west), with one trap placed in each quadrant to ensure systematic and spatially representative sampling. Fig 2 presents the trap locations.

The traps operated by using a light source to attract mosquitoes, which were then suctioned into a net by a motorized fan. Each morning at 6:00 AM, traps were retrieved, and the collection nets were labeled and refrigerated to kill the mosquitoes. The contents were then carefully transferred to petri dishes under controlled conditions to prevent sample loss due to wind or external disturbance. Collected mosquito specimens were stored in properly labeled Falcon tubes and submitted biweekly to the District Malaria Focal Person (MFP). The MFP was responsible for relabeling and packaging the samples for onward delivery to the Noguchi Memorial Institute for Medical Research.

For entomological data, we aggregated mosquito collections over the entire sampling period and compared species distributions between control and intervention groups. Differences in categorical distributions were assessed using the chi-square test.

**4. Feedback from local stakeholders.** Research members accompanied the spraying activities to observe operations and exchange views with field staff. After completing all field activities, a validation workshop was held in which the study was presented to stakeholders and open discussions were conducted to gather feedback on operational challenges and improvements. These inputs provided valuable insights into field implementation, but as they were not collected through a systematically designed survey and relied on stakeholders' subjective perspectives, they were treated as supplementary information in this study.

## Results

### Operational efficiency and data

To evaluate the effectiveness of drone-assisted versus conventional LSM mapping, simultaneous field mapping was conducted in the intervention sub-districts during September and October. Mapping on each sub-district was performed over

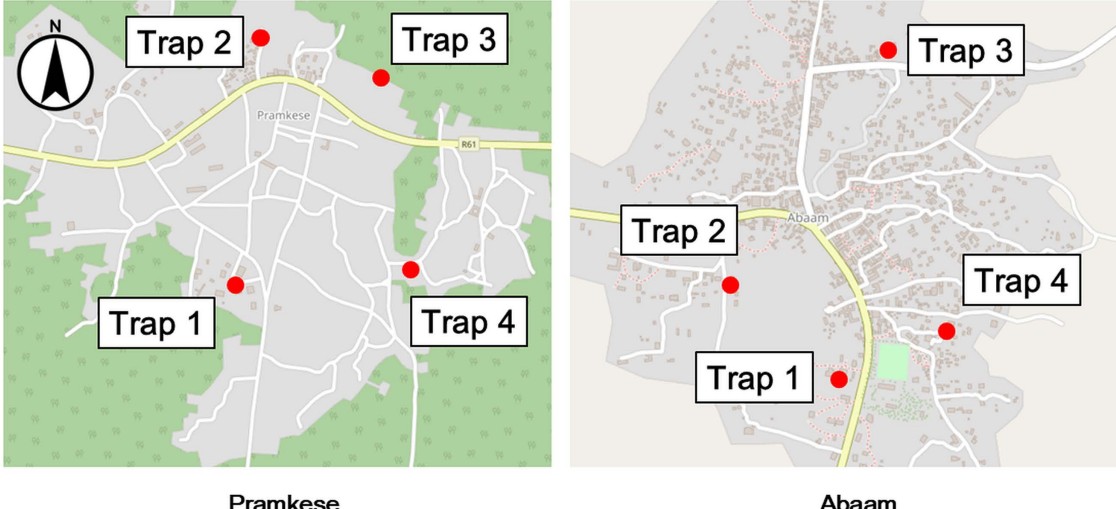

Pramkese                                      Abaam

**Fig 2. CDC light trap locations. Background map tiles © OpenStreetMap contributors, licensed under CC BY-SA 2.0. Additional annotations by the authors.**

a single day by both methods. Conventional mapping was executed by the same vector control contractor responsible for LSM in the control sub-districts. Table 3 summarizes the results of detected sites.

In every intervention sub-district, drone-based mapping identified a substantially higher number of waterbodies than the conventional approach. Detection ratios ranged from 1.83 to 4.68 times, with an overall average of 3.61 times more potential breeding sites identified by the drone method. Fig 3 visualizes the spatial distribution of identified sites. While conventional methods concentrate detections in central areas, drone-based mapping achieved wider geographic coverage, including peripheral and harder-to-reach zones. Furthermore, drone-mapped sites encompassed those identified through conventional scouting, suggesting that drone-based detection may serve as a viable alternative or even a superior substitute.

To quantify worker efficiency, we recorded the number of personnel and number of days required to implement each LSM process in both intervention and control sub-districts. These records were summarized in Tables 4 and 5. In the intervention sub-districts, aerial mapping, waterbody detection, and AI-based risk classification were performed the day before spraying and classified under the "Mapping" process. In the control sub-districts, mapping and spraying were conducted on the same day as part of a unified process. Total person-days were calculated for each sub-district as the sum of the personnel involved multiplied by the number of workdays.

To ensure comparability across sub-districts of different sizes, person-days were normalized per unit area for each sub-district. Table 6 presents the results of this comparison. Even after adjusting for area, the drone-based intervention method required less workers. On average, the normalized person-day per square kilometer in the intervention sub-districts was approximately 50% of that required in the control sub-districts. Based on the average values of TPUDA presented in Table 6, the labor cost margin obtained through the proposed approach was calculated using the following equation and summarized as Fig 4 with operation area on the x-axis:

$$M = R \cdot S \cdot (E_c - E_p) \qquad (2)$$

where $M$ denotes the worker cost margin, $R$ is the daily wage per worker, $S$ is the operation area, $E_c$ is the person-days per unit area for conventional manual LSM, and $E_p$ is the person-days per unit area for the proposed method. The value of parameter $R$ was derived from the income survey of public sector employees conducted by the Ghana Statistical Service (GSS) in 2022 [41]. In that year, the average net monthly salary was GHS 2,594. Accounting for an assumed 30% increase in 2023 and a further 23% increase in 2024, the adjusted monthly salary amounts to GHS 4,148. When divided by 20 working days, this corresponds to the daily wage of GHS 207.4.

Fig 5 further illustrates the number of operations required to recover the initial cost of drone procurement, using the labor cost margin defined above. The x-axis represents the initial drone cost, and each plot varies by operation area.

From the linear trend observed in Fig 4, it can be inferred that as the scale of operations increases, the use of drones leads to greater savings in labor costs. This suggests that drone based LSM may enable the implementation of larger scale interventions that would have been difficult to achieve under previous budget constraints. Fig 5 further illustrates the number of operations required to recover the initial cost of drone procurement using the estimated margin of labor cost

**Table 3. Waterbody sites detection result comparison.**

| Sub-District | Number of Sites by Drone | Number of Sites by Manual Scouting | Drone/Manual ratio | Operation Month |
|---|---|---|---|---|
| Nkwantanan | 154 | 34 | 4.53 | September |
| Abaam | 131 | 31 | 4.23 | September |
| Takyiman | 119 | 65 | 1.83 | October |
| Asuom | 290 | 62 | 4.68 | October |
| Total | 694 | 192 | 3.61 | – |

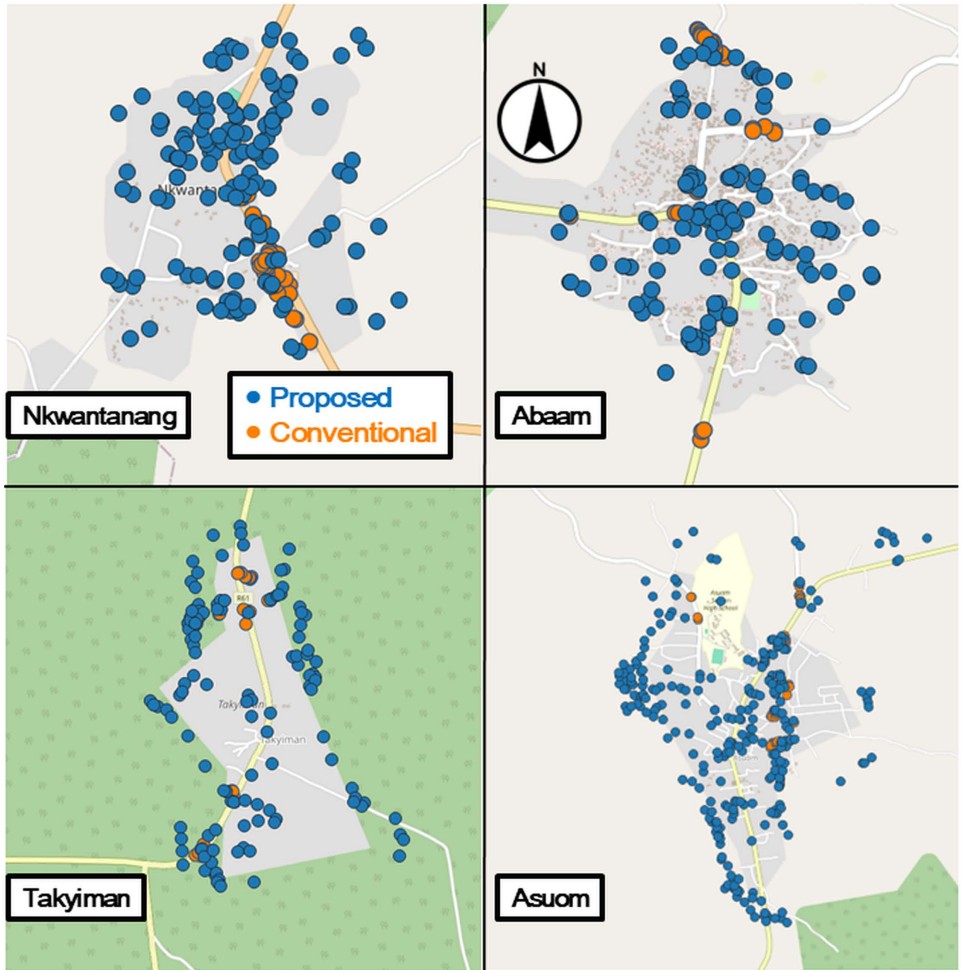

**Fig 3. Found sites distribution. Background map tiles © OpenStreetMap contributors, licensed under CC BY-SA 2.0. Additional annotations by the authors.**

Table 4. Summary of worker input in intervention.

| Sub-District | Mapping Workers | Mapping Days | Spraying Workers | Spraying Days | Total Person-Days |
|---|---|---|---|---|---|
| Nkwantanang | 2 | 1 | 3 | 1 | 5 |
| Abaam | 2 | 1 | 3 | 1 | 5 |
| Takyiman | 2 | 1 | 3 | 1 | 5 |
| Asuom | 2 | 3 | 3 | 2 | 12 |

savings. The drone used in this study costs approximately 75,000 GHC, and for an LSM operation covering an area of about 10 km², the initial investment is projected to be recovered after approximately ten operational cycles. Assuming one application per month, the cost of drone procurement could therefore be recovered within one year. Although the lifespan of drones is not standardized, even under the conservative assumption of three to five years, recovering the initial investment within the first year indicates that drone adoption represents a practical and economically advantageous strategy for expanding LSM activities.

**Table 5. Summary of worker input in control.**

| Sub-District | Mapping and Spraying Workers | Mapping and Spraying Days | Total Person-Days |
|---|---|---|---|
| Abodom | 6 | 2 | 12 |
| Subi | 6 | 1 | 6 |
| Pramkese | 6 | 2 | 12 |
| Kade | 8 | 6 | 48 |

**Table 6. Summary of worker input comparison.**

| Control/ Intervention | Sub-district | Total Person-Days | TPDUA [km$^{-2}$] |
|---|---|---|---|
| Intervention | Nkwantanang | 5 | 5.32 |
| Intervention | Abaam | 5 | 4.85 |
| Intervention | Takyiman | 5 | 2.82 |
| Intervention | Asuom | 12 | 2.99 |
| Intervention | Intervention Average | 6.75 | 4.00 |
| Control | Abodom | 12 | 12.24 |
| Control | Subi | 6 | 5.41 |
| Control | Pramkese | 12 | 7.74 |
| Control | Kade | 48 | 6.72 |
| Control | Control Average | 19.5 | 8.03 |

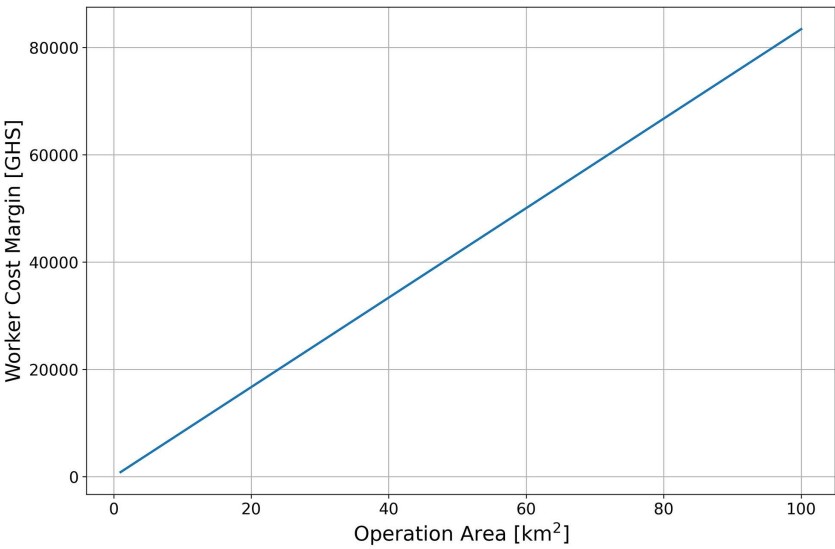

**Fig 4. Worker cost savings per operation area scale.**

For operational efficiency measured as total person-days per unit area (TPDUA), the median was significantly lower in the intervention sites (3.92 [IQR 2.91–5.09]) compared to the control sites (7.23 [IQR 6.07–9.99]). This difference was statistically significant (Mann–Whitney U = 0.0, p = 0.029).

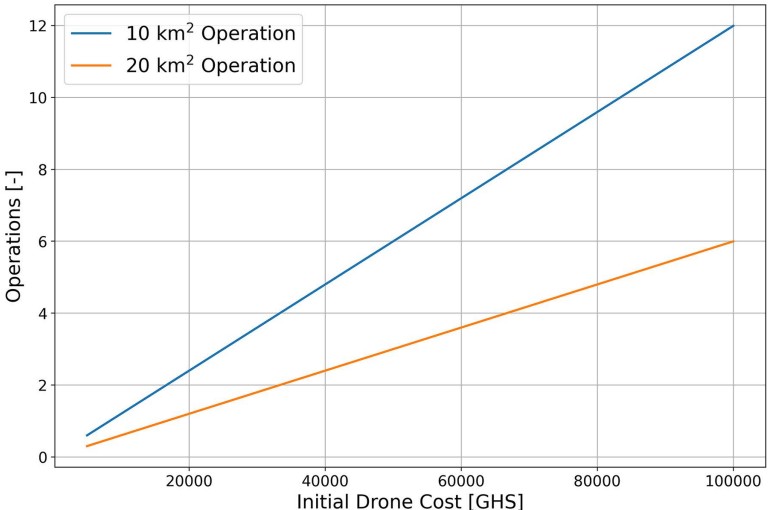

**Fig 5. Operations needed to recover drone costs through Worker cost savings alone.**

Table 7 compares larvicide consumption between the intervention and control sub-districts. In the intervention arm, spraying was limited to sites classified as high-risk by the AI model. The high-risk classification ratios were 0.557 (Nkwantanan), 0.592 (Abaam), 0.747 (Takyiman), and 0.585 (Asuom), and were used to derive the number of treated waterbodies. In contrast, in the control sub-districts, all detected waterbodies were treated without risk prioritization.

Interestingly, even though more breeding sites were treated in the intervention sub-districts, the number of treated sites per larvicide pack unit was significantly higher. The drone-assisted intervention achieved 169 treated sites per unit of larvicide, compared to 60 in the control, representing a two-fold improvement in larvicide efficiency. It should be noted that larvicide allocation was predetermined based on sub-district area, and the research team was not involved in dosing decisions; these were managed independently by the spraying contractor.

For larvicide use efficiency, measured as the number of sites treated per pack of larvicide, the intervention sites achieved substantially higher values (median 174.0 [IQR 158.0–180.0]) compared to the control sites (median 49.0 [IQR 42.0–77.5]). This difference was statistically significant (Mann–Whitney U = 0.0, p = 0.029).

**Table 7. Summary of larvicide usage.**

| Control/ Intervention | Sub-district | Number of Found Sites | Sprayed Sites | Larvicide Packs per Month | Sites per Larvicide Packs |
|---|---|---|---|---|---|
| Intervention | Nkwantanang | 131 | 73 | 0.5 | 146 |
| Intervention | Abaam | 154 | 91 | 0.5 | 182 |
| Intervention | Takyiman | 119 | 89 | 0.5 | 178 |
| Intervention | Asuom | 290 | 170 | 1 | 170 |
| Intervention | Intervention Average | 174 | 106 | 0.63 | **169** |
| Control | Abodom | 18 | 18 | 0.5 | 36 |
| Control | Subi | 25 | 25 | 0.5 | 50 |
| Control | Pramkese | 24 | 24 | 0.5 | 48 |
| Control | Kade | 210 | 210 | 2 | 105 |
| Control | Control Average | 69 | 69 | 0.88 | **60** |

To validate the performance of AI-based risk classification, comprehensive entomological surveys were conducted in November in Takyiman, covering all waterbodies identified by LSM. Fig 6 presents the confusion matrix comparing AI risk prediction to ground-truth larval presence (positive if larvae observed; negative if absent).

The classification metrics were as follows: Accuracy = 0.70, Precision = 0.40, Recall = 0.875. The high recall indicates the AI system rarely misses actual breeding sites, which is desirable in public health applications where false negatives carry higher risk. However, the relatively low precision suggests room for improvement in avoiding unnecessary treatment of non-productive sites.

Given the complexity of malaria transmission and the need to minimize missed breeding sites during early AI deployment phases, a recall-oriented model aligns well with operational priorities. In this context, the AI system's current performance represents a practical and risk-averse implementation pathway.

## Epidemiological data

Epidemiological data were collected based on OPD malaria case records from eight sub-districts. The analysis covered the entire study period, with specific comparisons focused on the four months between September 16 and January 17. Monthly malaria case counts were compared across years, with trend analysis incorporating historical data from 2022 and 2023. Fig 5 illustrates the time series of OPD malaria cases, separated into the intervention group (A) and control group (B). "NY" in Fig 7 denotes the subsequent calendar next year.

Both intervention and control sub-districts showed a consistent seasonal trend, with malaria cases peaking between May and July and gradually declining through January. Notably, in 2022 and 2023, years when LSM was not implemented in the intervention sub-districts, these sub-districts still tended to report fewer cases than the control sub-districts, where LSM had already been deployed. This suggests a historical tendency for lower malaria burden in the intervention districts. However, in 2024, the pattern diverged. The intervention sub-districts reported 1,312 cases at the July peak, a number comparable to the control group's 1,389 cases, indicating an unusually high malaria burden compared to previous years.

Fig 8 presents the 2024 time-series trends in malaria case counts across both groups. Vertical dashed lines indicate the month of LSM initiation and completion in each group. Panel (A) displays raw OPD case counts, while panel (B) shows population-adjusted case rates, calculated by dividing monthly case counts by the total population for each sub-district (as reported in Table 1).

Both groups reached a peak in July, followed by a gradual decline until September, then relatively stable case numbers through December, and a sharp drop in January. While raw case numbers remained similar between groups, the population-adjusted rates were higher in the intervention group. LSM was initiated two months earlier in the control sub-districts (July) than in the intervention sub-districts (September). Overall, the temporal patterns of case numbers in both groups appeared broadly consistent with expected seasonal dynamics.

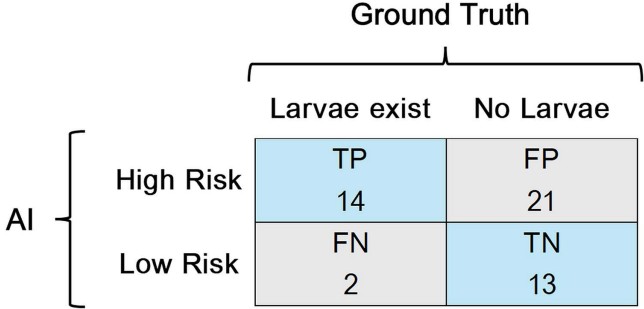

**Fig 6. AI Estimation and ground truth in November Takyiman.**

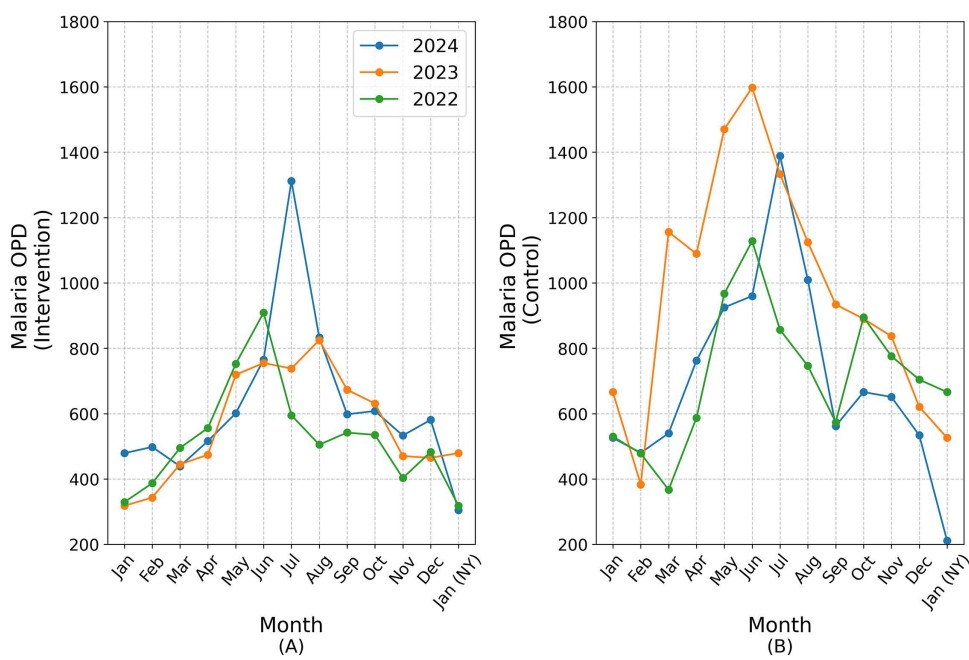

**Fig 7. Malaria OPD time series.**

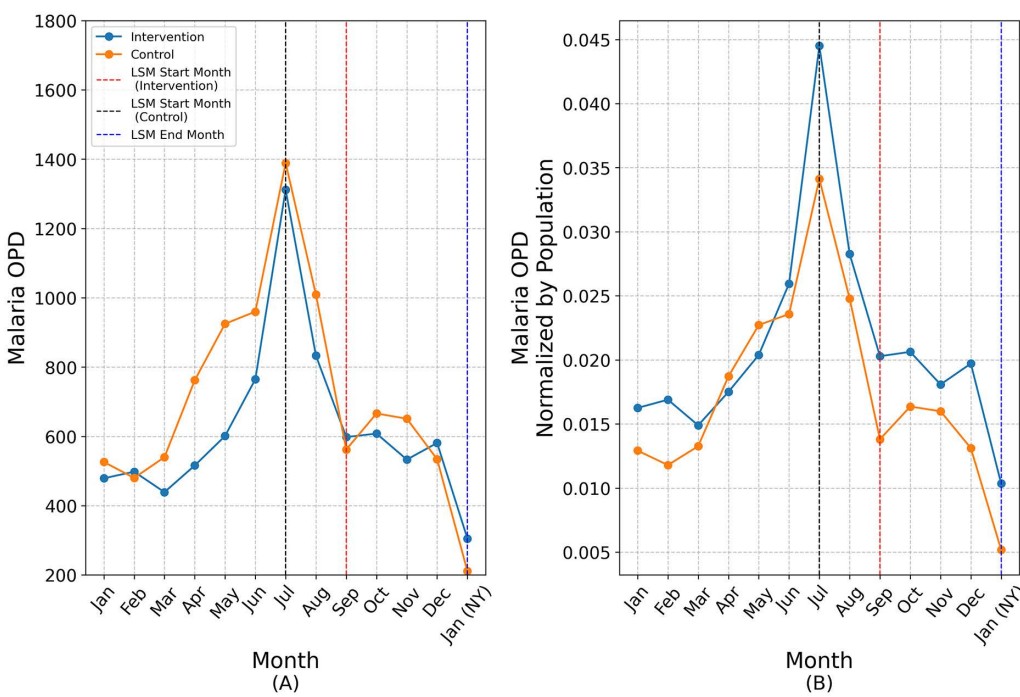

**Fig 8. Comparison of intervention and control in 2024.**

## Entomological data

A total of 964 insects were captured during the study period, of which 783 (81.2%) were mosquitoes. Among these, 430 (45.1%) mosquitoes were collected in Pramkese and 353 (54.9%) in Abaam. Regarding sex distribution, males accounted for 192 (24.5%) and females 591 (75.5%). Morphological identification revealed the presence of four mosquito genera, with substantial variation in their relative abundances. *Culex* was overwhelmingly dominant, comprising 90.8% of the total mosquito catch. *Aedes* and *Anopheles* represented 4.73% and 4.34%, respectively, while *Mansonia* was least frequent, at just 0.13%. An additional 181 non-target insects, primarily moths, were also collected. Fig 9 summarizes the overall insect capture results.

The survey period (September to January) may have coincided with climatic conditions (e.g., rainfall patterns and temperature fluctuations) unfavorable for *Anopheles* and *Aedes*. As mosquito abundance often varies by season and ecological niche, the dominance of *Culex* suggests that the environmental conditions during the study were more supportive of this genus. This pattern may also indicate that the study areas lacked habitats suitable for *Anopheles*, the primary malaria vector. Moreover, the abundance of *Culex* suggests the presence of multiple breeding sites such as polluted and stagnant water bodies, which may raise broader public health concerns.

Although the low number of *Anopheles* specimens precludes a reliable time-series evaluation of the LSM intervention's effect on malaria vectors specifically, there remains value in analyzing the temporal dynamics of all mosquito species collectively. Fig 10 presents the integrated time-series data on mosquito captures. During the first five weeks, Pramkese consistently recorded more mosquitoes than Abaam, with particularly notable differences in Week 1 (84 vs. 45) and Week 2 (77 vs. 29). However, in Week 6, this trend reversed sharply, with Abaam recording 54 mosquitoes compared to 16 in Pramkese. In Weeks 8 and 9, Abaam recorded zero captures, while Pramkese continued to report moderate levels (33 and 44, respectively). A striking reversal occurred in Week 11, when Abaam recorded a sharp increase (82 mosquitoes), while Pramkese saw a decline (17 mosquitoes). In the final week, counts declined in both sub-districts, with Abaam at 10 and Pramkese at 31.

These temporal fluctuations reflect the complex and dynamic nature of mosquito populations in both study areas. The temporary absence and subsequent resurgence of mosquitoes in Abaam may suggest a short-term suppressive effect of

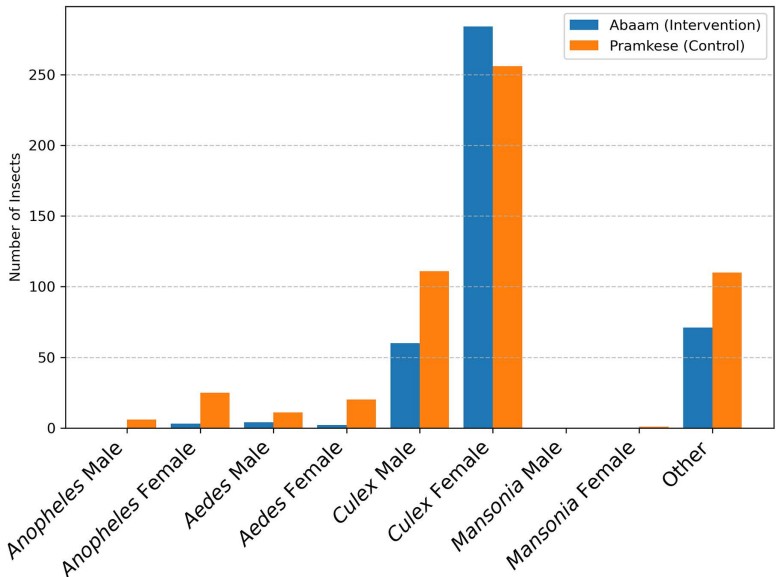

**Fig 9. Results of entomological data collection.**

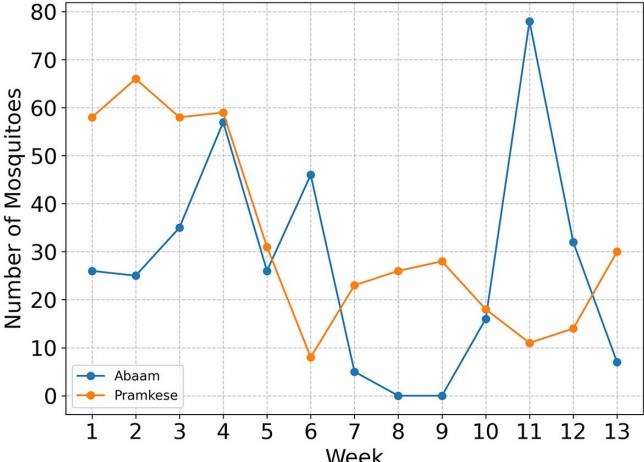

**Fig 10. Time-series comparison of captured mosquito.**

the intervention, while the sharp increase in Week 11 indicates that other environmental or ecological factors may have also influenced mosquito abundance. The lower total mosquito count in the intervention site Abaam compared to the control site Pramkese (353 vs. 430) may provide preliminary evidence of intervention impact. However, no clear qualitative differences in total abundance or temporal trends were observed between the two sub-districts. Therefore, no distinct differences in *Anopheles* control performance between LSM strategies were confirmed in this study.

To evaluate differences in mosquito species distributions between the control and intervention sites, we aggregated capture data over the study period and constructed the contingency table shown in Table 8. Since only one *Mansonia* specimen was collected, this genus was excluded from the analysis. Based on this contingency table, a chi-square test was performed. The results indicated a significant difference in species distribution between Abaam and Pramkese ($\chi^2(2)$ = 33.63, p < 0.001). Specifically, *Anopheles* was more prevalent in Pramkese, whereas *Culex* predominated in Abaam.

### Qualitative insights from local stakeholders

A validation workshop was held in January 2025 at the Kwaebibirem Municipal Health Directorate, organized by the Ghana Health Service. Participants included clinicians, health administrators, and local stakeholders from both intervention and control sub-districts, who provided feedback to complement the quantitative findings of this study. Attendance was voluntary and documented with signed consent.

During the validation workshop, several clinicians reported a noticeable decline in malaria cases in December and January, particularly in facilities such as Abaam and Nkwantanang. These perspectives represent subjective observations from participants and were included as supplementary context to support the quantitative outpatient records. Field teams

**Table 8. Contingency table of mosquito.**

| Mosquito | Abaam (Intervention) | Pramkese (Control) | Total |
|---|---|---|---|
| Anopheles | 3 | 31 | 34 |
| Aedes | 6 | 31 | 37 |
| Culex | 344 | 367 | 711 |
| Total | 353 | 429 | 782 |

responsible for larvicide application also reported that drone-based mapping improved the accuracy of waterbody identification compared to manual methods. Despite identifying more waterbodies, teams noted that total larvicide use declined, suggesting improved efficiency.

## Discussion

This study demonstrated that integrating drone and AI technologies into LSM offers multiple operational and systemic advantages compared to conventional manual methods. Drone-based mapping enabled broader and more comprehensive coverage of potential breeding sites, and when combined with AI-driven risk classification, contributed to more efficient use of human resources and larvicides. Although the study was conducted mainly during the dry season (September–December), drone surveys still detected an increased number of waterbodies, suggesting that performance would be even stronger in the rainy season. We acknowledge that drone operations may be limited by adverse weather and that habitats fully concealed under canopy or structures cannot be detected; in such cases, complementary ground-based surveys remain necessary.

The AI model used in this study achieved a high recall, indicating its ability to detect true mosquito breeding sites without overlooking them. This is a critical feature for public health applications. This supports the potential of AI integration to enhance efficiency in LSM operations. However, the model's relatively low precision may be attributed to limitations in the number and representativeness of training data, as well as the constraints of using only visible-spectrum drone imagery. The moderate accuracy achieved despite training on data collected outside the intervention area suggests that the model captured generalizable patterns. Nevertheless, future model improvement would benefit from incorporating locally sourced data to reflect sub-district-specific ecological conditions more accurately.

A key limitation of this study was the low number of *Anopheles* collected, which made robust statistical evaluation of the intervention's direct impact on malaria transmission challenging. This likely reflects the short intervention window, seasonal declines in transmission, and the absence of positivity rate data. *Culex mosquitoes* predominated; while they are known vectors of other diseases globally, no such transmission has been reported in Ghana [42]. Their reduction may nonetheless bring ancillary benefits by reducing nuisance biting. The study was further limited by the absence of systematic larval surveys and precise measurements of treated waterbody areas, preventing area-based efficiency analysis. Larval AI performance was supported by limited larval checks in November, but outcomes were mainly assessed through adult mosquito abundance and malaria case records. A chi-square test indicated species distribution differences, especially for *Anopheles*, but sparse sampling means this result should be regarded as supportive rather than conclusive. Interpretation of epidemiological outcomes was constrained by the short intervention period, overlapping with seasonal declines, and potential confounding from weather, concurrent interventions, and reporting differences. Taken together, observed reductions in malaria cases are more plausibly explained by seasonal variation than by the intervention itself, though importantly malaria outcomes were not worse despite reduced larvicide use and labor inputs.

Feedback from field staff and health workers provided important operational insights. Although more waterbodies were detected in the intervention sites, total larvicide use declined, indicating that larvicide volume alone is an insufficient metric for evaluating efficiency. Quantitative results supported this observation: TPDUA was significantly lower in the intervention sites than in the control sites (median 3.92 [IQR 2.91 to 5.09] versus 7.23 [IQR 6.07 to 9.99], Mann–Whitney U = 0.0, p = 0.029), demonstrating that the drone- and AI-assisted workflow achieved roughly twice the labor efficiency of the conventional method. Similarly, the number of sites treated per pack of larvicide was substantially higher in the intervention sites (median 174.0 [IQR 158.0 to 180.0] versus 49.0 [IQR 42.0 to 77.5], Mann–Whitney U = 0.0, p = 0.029), corresponding to about 2.8 times greater larvicide use efficiency. Taking them together, these findings indicate meaningful operational gains, although the small sample size (n = 4 per group) warrants cautious interpretation.

From an economic perspective, the findings of this study indicate that the introduction of drones and AI can potentially reduce labor costs by approximately half, even when implementing LSM at scales comparable to conventional methods.

Previous study on LSM cost structures have reported that labor accounts for about 58 percent of total costs [43], suggesting that the proposed approach could substantially alleviate this major cost component. The magnitude of labor cost savings is expected to increase as the operational area expands, which implies that larger LSM coverage may be achievable within the same budget. In addition, for operations of a scale like that examined here, the initial investment required for drone procurement could be recovered within roughly one year through labor cost savings alone, indicating a favorable cost-effectiveness profile over the long term. Larvicide constitutes another considerable portion of LSM costs, accounting for around 14 percent [43]. Improvements in spraying efficiency can therefore reduce larvicide consumption, leading to lower operational costs per cycle. Overall, the proposed method may enable LSM implementation with fewer personnel and reduced larvicide use while maintaining or lowering total budget requirements, making it a promising strategy for sustainable vector control in resource-constrained settings.

Several clinicians also reported sharper-than-expected reductions in malaria cases in intervention sub-districts, implying that precise targeting of high-risk sites may yield observable impact even within a short period. Community acceptance of drone flights was not formally assessed in this study. Nevertheless, informal observations made by clinicians and field teams indicated that community reactions varied across sites, largely reflecting differences in the extent of sensitization activities conducted prior to implementation. In Abaam, where sensitization was extensive, residents were welcoming and even assisted spray teams in identifying breeding sites. By contrast, in Nkwantanang and Takyiman, some residents expressed hesitation about larvicide application in open waterbodies used for household purposes. In Asuom, drone flights themselves were met with unease, as communities feared possible exposure of illegal mining activities. Although these reactions did not escalate into overt resistance or operational disruption, they highlight the need for systematic evaluation of community perceptions and acceptance, an essential factor for the sustainability and scalability of drone- and AI-assisted LSM interventions. Because these insights were based on subjective impressions rather than structured data collection, they should be interpreted cautiously. Future studies should incorporate formal surveys or mixed-methods assessments to capture community perspectives alongside quantitative outcomes.

To support community understanding and cooperation, community engagement activities were conducted in each sub-district prior to implementation. Explanatory meetings with community elders were held to secure approval for the purpose, procedures, and flight plan of the LSM operation, and additional announcements were disseminated through local public speaker systems. Information sharing continued throughout the intervention period with support from the district NMEP office. Although some residents expressed hesitation or unease, no conflicts or operational disturbances were reported, and LSM activities proceeded smoothly, likely reflecting both prior community experience with LSM interventions and the effectiveness of early communication efforts.

Resident concerns regarding drone use centered on noise, crash risk, and privacy. With respect to noise, the sound of multirotor propellers may be perceived as unpleasant. However, the Mavic 3M was operated at approximately 100 m altitude, which substantially reduced sound pressure at ground level. Previous studies report that drones of similar size generate approximately 40–50 dB directly beneath the aircraft at about 120 m altitude [44]. These values fall within or below the World Health Organization daytime guideline thresholds for transportation noise, which are 53 dB for road traffic and 45 dB for aircraft [45]. Consistent with these expectations, no noise-related complaints were recorded. Even so, thresholds for noise annoyance vary across individuals and cultural contexts, indicating the need for future work that incorporates acoustic measurements and resident surveys. Crash risk cannot be eliminated entirely, but several mitigation measures were implemented. These included mandatory helmet use for field teams, restricting flights to an area of 300 m by 300 m that could be visually monitored, operating only with fully charged batteries, and avoiding adverse weather conditions. Regarding privacy, flight permission was obtained under regulations that prohibit secondary use of collected imagery. At 100 m altitude, the ground sampling distance was approximately 3 cm per pixel, which allows recognition of human presence but makes personal identification highly unlikely. Although household structures were visible, which introduces a residual risk of inferring household characteristics, such information can support malaria control activities such as

planning for indoor residual spraying. Maintaining an appropriate balance between public health utility and privacy protection remains essential.

In this study, the detection of waterbodies was conducted manually using drone images, with only minimal support from image-recognition AI. This decision reflects the current state of technology: while previous studies have demonstrated the potential of automated waterbody detection, such models are not yet standardized or consistently reliable for operational deployment, and their performance is fundamentally limited by reliance on human-annotated training data. Replacing this step with manual identification therefore did not compromise the validity of our approach. The key novelty of this study lies instead in demonstrating, for the first time, the operational value of integrating AI-based risk classification into a three-step pipeline of drone imagery, waterbody identification, and larvicide implementation. Importantly, our findings highlight that this integration improved efficiency even when the detection step remained manual, underscoring its practical applicability under current technological constraints. Future integration of lightweight computer vision models could further automate this step, particularly if optimized for resource-constrained field settings. Similarly, while this study focused on combining drone-based mapping with manual spraying, incorporating spraying drones represents a promising direction toward more automated LSM. At the same time, integrating large and heavy spraying drones into urban environments while ensuring safe operation remains a critical challenge. Future research should therefore assess the feasibility of spraying drones under appropriate ecological and operational conditions, with particular attention to safety and community acceptance. Developing these technological components presents significant challenges but holds substantial potential for improving the scalability, efficiency, and sustainability of digital LSM. Based on these findings, the following points should be prioritized in future research:

- Longitudinal studies covering full malaria transmission seasons

- Expanded entomological surveillance focusing on *Anopheles* species

- Economic evaluations incorporating health outcomes (e.g., cost-effectiveness, DALYs averted)

- Policy research on shifting performance metrics and incentive structures away from resource usage and toward operational and health impact

## Conclusion

This study provides evidence that drone and AI-assisted LSM can improve operational efficiency without undermining vector control effectiveness. While epidemiological effects remain inconclusive, operational gains suggest strong potential for scalable implementation. Longer-term studies are warranted.

## Acknowledgments

We gratefully acknowledge the collaboration of the Ghana National Malaria Elimination Program and the Ghana Health Service for their guidance and logistical support throughout the implementation. We thank Dr. Gaku Masuda of Tokyo Women's Medical University, Section of Global Health, Department of Hygiene and Public Health, School of Medicine, for his valuable advice on data interpretation and manuscript development. We also acknowledge the Takemi Program for International Health for its support during the writing of the paper.

## Author contributions

**Conceptualization:** Masaki Umeda.

**Data curation:** Samuel Dadzie.

**Formal analysis:** Francis A. Adzei.

**Investigation:** Godfred A. Bokpin, Francis A. Adzei.

**Software:** Juhoe Kim.

**Supervision:** Godfred A. Bokpin.

**Visualization:** Juhoe Kim.

**Writing – original draft:** Godfred A. Bokpin.

**Writing – review & editing:** Juhoe Kim.

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
