## [Decision Letter · Decision Letter 0]

21 Jul 2025

Dear Dr. Kim,

Thank you for submitting your manuscript to PLOS ONE. After careful consideration, we feel that it has merit but does not fully meet PLOS ONE’s publication criteria as it currently stands. Therefore, we invite you to submit a revised version of the manuscript that addresses the points raised during the review process.

We look forward to receiving your revised manuscript.

Kind regards,

Himmat Singh

Academic Editor

PLOS ONE

**Journal Requirements:**

1. When submitting your revision, we need you to address these additional requirements. Please ensure that your manuscript meets PLOS ONE's style requirements, including those for file naming. The PLOS ONE style templates can be found at https://journals.plos.org/plosone/s/file?id=wjVg/PLOSOne_formatting_sample_main_body.pdf and https://journals.plos.org/plosone/s/file?id=ba62/PLOSOne_formatting_sample_title_authors_affiliations.pdf 2. Please include a complete copy of PLOS’ questionnaire on inclusivity in global research in your revised manuscript. Our policy for research in this area aims to improve transparency in the reporting of research performed outside of researchers’ own country or community. The policy applies to researchers who have travelled to a different country to conduct research, research with Indigenous populations or their lands, and research on cultural artefacts. The questionnaire can also be requested at the journal’s discretion for any other submissions, even if these conditions are not met.  Please find more information on the policy and a link to download a blank copy of the questionnaire here: https://journals.plos.org/plosone/s/best-practices-in-research-reporting. Please upload a completed version of your questionnaire as Supporting Information when you resubmit your manuscript. 3. Please note that PLOS ONE has specific guidelines on code sharing for submissions in which author-generated code underpins the findings in the manuscript. In these cases, we expect all author-generated code to be made available without restrictions upon publication of the work. Please review our guidelines at https://journals.plos.org/plosone/s/materials-and-software-sharing#loc-sharing-code and ensure that your code is shared in a way that follows best practice and facilitates reproducibility and reuse. 4. Thank you for stating in your Funding Statement: This study was supported by funding from the Japan International Cooperation Agency (JICA) and the Japan Science and Technology Agency (JST) under the framework of international technical cooperation. The funding was administered through SORA Technology Ltd., which coordinated the implementation activities. The funders had no direct role in the decision to submit the manuscript for publication.  Please provide an amended statement that declares *all* the funding or sources of support (whether external or internal to your organization) received during this study, as detailed online in our guide for authors at http://journals.plos.org/plosone/s/submit-now.  Please also include the statement “There was no additional external funding received for this study.” in your updated Funding Statement. Please include your amended Funding Statement within your cover letter. We will change the online submission form on your behalf. 5. Thank you for stating the following in the Competing Interests section: This study was conducted with funding administered through SORA Technology Ltd., to which authors Masaki Umeda and Juhoe Kim are affiliated. Authors Godfred A. Bokpin, and Francis A. Adzei received research support through this funding mechanism. Authors affiliated with SORA Technology were involved in the study design, implementation, and manuscript preparation. All other authors declare no competing interests.  We note that one or more of the authors are employed by a commercial company.  a. Please provide an amended Funding Statement declaring this commercial affiliation, as well as a statement regarding the Role of Funders in your study. If the funding organization did not play a role in the study design, data collection and analysis, decision to publish, or preparation of the manuscript and only provided financial support in the form of authors' salaries and/or research materials, please review your statements relating to the author contributions, and ensure you have specifically and accurately indicated the role(s) that these authors had in your study. You can update author roles in the Author Contributions section of the online submission form. Please also include the following statement within your amended Funding Statement. “The funder provided support in the form of salaries for authors [insert relevant initials], but did not have any additional role in the study design, data collection and analysis, decision to publish, or preparation of the manuscript. The specific roles of these authors are articulated in the ‘author contributions’ section.”If your commercial affiliation did play a role in your study, please state and explain this role within your updated Funding Statement.  b. Please also provide an updated Competing Interests Statement declaring this commercial affiliation along with any other relevant declarations relating to employment, consultancy, patents, products in development, or marketed products, etc.   Within your Competing Interests Statement, please confirm that this commercial affiliation does not alter your adherence to all PLOS ONE policies on sharing data and materials by including the following statement: "This does not alter our adherence to  PLOS ONE policies on sharing data and materials.” (as detailed online in our guide for authors http://journals.plos.org/plosone/s/competing-interests) . If this adherence statement is not accurate and  there are restrictions on sharing of data and/or materials, please state these. Please note that we cannot proceed with consideration of your article until this information has been declared. Please include both an updated Funding Statement and Competing Interests Statement in your cover letter. We will change the online submission form on your behalf. 6. We note that your Data Availability Statement is currently as follows: All relevant data are within the manuscript and its Supporting Information files. Please confirm at this time whether or not your submission contains all raw data required to replicate the results of your study. Authors must share the “minimal data set” for their submission. PLOS defines the minimal data set to consist of the data required to replicate all study findings reported in the article, as well as related metadata and methods (https://journals.plos.org/plosone/s/data-availability#loc-minimal-data-set-definition). For example, authors should submit the following data: - The values behind the means, standard deviations and other measures reported;- The values used to build graphs;- The points extracted from images for analysis. Authors do not need to submit their entire data set if only a portion of the data was used in the reported study. If your submission does not contain these data, please either upload them as Supporting Information files or deposit them to a stable, public repository and provide us with the relevant URLs, DOIs, or accession numbers. For a list of recommended repositories, please see https://journals.plos.org/plosone/s/recommended-repositories. If there are ethical or legal restrictions on sharing a de-identified data set, please explain them in detail (e.g., data contain potentially sensitive information, data are owned by a third-party organization, etc.) and who has imposed them (e.g., an ethics committee). Please also provide contact information for a data access committee, ethics committee, or other institutional body to which data requests may be sent. If data are owned by a third party, please indicate how others may request data access. 7. When completing the data availability statement of the submission form, you indicated that you will make your data available on acceptance. We strongly recommend all authors decide on a data sharing plan before acceptance, as the process can be lengthy and hold up publication timelines. Please note that, though access restrictions are acceptable now, your entire data will need to be made freely accessible if your manuscript is accepted for publication. This policy applies to all data except where public deposition would breach compliance with the protocol approved by your research ethics board. If you are unable to adhere to our open data policy, please kindly revise your statement to explain your reasoning and we will seek the editor's input on an exemption. Please be assured that, once you have provided your new statement, the assessment of your exemption will not hold up the peer review process. 8. Please include your full ethics statement in the ‘Methods’ section of your manuscript file. In your statement, please include the full name of the IRB or ethics committee who approved or waived your study, as well as whether or not you obtained informed written or verbal consent. If consent was waived for your study, please include this information in your statement as well. 9. We note that Figures 2 and 3 in your submission contain map/satellite images which may be copyrighted. All PLOS content is published under the Creative Commons Attribution License (CC BY 4.0), which means that the manuscript, images, and Supporting Information files will be freely available online, and any third party is permitted to access, download, copy, distribute, and use these materials in any way, even commercially, with proper attribution. For these reasons, we cannot publish previously copyrighted maps or satellite images created using proprietary data, such as Google software (Google Maps, Street View, and Earth). For more information, see our copyright guidelines: http://journals.plos.org/plosone/s/licenses-and-copyright. We require you to either present written permission from the copyright holder to publish these figures specifically under the CC BY 4.0 license, or remove the figures from your submission: a. You may seek permission from the original copyright holder of Figures 2 and 3  to publish the content specifically under the CC BY 4.0 license.   We recommend that you contact the original copyright holder with the Content Permission Form (http://journals.plos.org/plosone/s/file?id=7c09/content-permission-form.pdf) and the following text:“I request permission for the open-access journal PLOS ONE to publish XXX under the Creative Commons Attribution License (CCAL) CC BY 4.0 (http://creativecommons.org/licenses/by/4.0/). Please be aware that this license allows unrestricted use and distribution, even commercially, by third parties. Please reply and provide explicit written permission to publish XXX under a CC BY license and complete the attached form.” Please upload the completed Content Permission Form or other proof of granted permissions as an "Other" file with your submission. In the figure caption of the copyrighted figure, please include the following text: “Reprinted from [ref] under a CC BY license, with permission from [name of publisher], original copyright [original copyright year].” b. If you are unable to obtain permission from the original copyright holder to publish these figures under the CC BY 4.0 license or if the copyright holder’s requirements are incompatible with the CC BY 4.0 license, please either i) remove the figure or ii) supply a replacement figure that complies with the CC BY 4.0 license. Please check copyright information on all replacement figures and update the figure caption with source information. If applicable, please specify in the figure caption text when a figure is similar but not identical to the original image and is therefore for illustrative purposes only.The following resources for replacing copyrighted map figures may be helpful: USGS National Map Viewer (public domain): http://viewer.nationalmap.gov/viewer/The Gateway to Astronaut Photography of Earth (public domain): http://eol.jsc.nasa.gov/sseop/clickmap/Maps at the CIA (public domain): https://www.cia.gov/library/publications/the-world-factbook/index.html and https://www.cia.gov/library/publications/cia-maps-publications/index.htmlNASA Earth Observatory (public domain): http://earthobservatory.nasa.gov/Landsat: http://landsat.visibleearth.nasa.gov/USGS EROS (Earth Resources Observatory and Science (EROS) Center) (public domain): http://eros.usgs.gov/#Natural Earth (public domain): http://www.naturalearthdata.com/ 10. We note that Figure 1 in your submission contain copyrighted images. All PLOS content is published under the Creative Commons Attribution License (CC BY 4.0), which means that the manuscript, images, and Supporting Information files will be freely available online, and any third party is permitted to access, download, copy, distribute, and use these materials in any way, even commercially, with proper attribution. For more information, see our copyright guidelines: http://journals.plos.org/plosone/s/licenses-and-copyright. We require you to either present written permission from the copyright holder to publish these figures specifically under the CC BY 4.0 license, or remove the figures from your submission: a. You may seek permission from the original copyright holder of Figure 1 to publish the content specifically under the CC BY 4.0 license.  We recommend that you contact the original copyright holder with the Content Permission Form (http://journals.plos.org/plosone/s/file?id=7c09/content-permission-form.pdf) and the following text:“I request permission for the open-access journal PLOS ONE to publish XXX under the Creative Commons Attribution License (CCAL) CC BY 4.0 (http://creativecommons.org/licenses/by/4.0/). Please be aware that this license allows unrestricted use and distribution, even commercially, by third parties. Please reply and provide explicit written permission to publish XXX under a CC BY license and complete the attached form.” Please upload the completed Content Permission Form or other proof of granted permissions as an "Other" file with your submission.  In the figure caption of the copyrighted figure, please include the following text: “Reprinted from [ref] under a CC BY license, with permission from [name of publisher], original copyright [original copyright year].” b. If you are unable to obtain permission from the original copyright holder to publish these figures under the CC BY 4.0 license or if the copyright holder’s requirements are incompatible with the CC BY 4.0 license, please either i) remove the figure or ii) supply a replacement figure that complies with the CC BY 4.0 license. Please check copyright information on all replacement figures and update the figure caption with source information. If applicable, please specify in the figure caption text when a figure is similar but not identical to the original image and is therefore for illustrative purposes only. 11. If the reviewer comments include a recommendation to cite specific previously published works, please review and evaluate these publications to determine whether they are relevant and should be cited. There is no requirement to cite these works unless the editor has indicated otherwise. 

**Additional Editor Comments:**

The MS need to be revised in view of the comments given by both the reviewer as major revision .

The methods of AI ML learning needs explanation as per the efficiency for detection of waterbodies, other breeding sites , the distance of drone from the waterbodies how larval risk was calculated, how correctly the mapping was done, what larvicide and how it was put

Reviewers' comments:

**Comments to the Author**

1. Is the manuscript technically sound, and do the data support the conclusions?

Reviewer #1: Partly

Reviewer #2: Yes

2. Has the statistical analysis been performed appropriately and rigorously?

Reviewer #1: No

Reviewer #2: N/A

3. Have the authors made all data underlying the findings in their manuscript fully available?

Reviewer #1: Yes

Reviewer #2: Yes

4. Is the manuscript presented in an intelligible fashion and written in standard English?

Reviewer #1: Yes

Reviewer #2: Yes

**Reviewer #1:**  Overall, this is an interesting premise. This manuscript presents field evaluation of a drone- and AI-assisted approach to larval source management (LSM) in Ghana. The study compares this tech-enhanced method to conventional manual LSM across eight sub-districts and assesses differences in operational efficiency, resource use, and malaria trends. However, I am thoroughly disappointed in the detail and framing of everything presented here. This paper still needs extensive improvements on transparency of methods and some improved data analysis. The authors sorely need to address their overreach of conclusions and do a better job of narratively incorporating existing literature on their topic in the discussion. Some clarifications are needed from the authors before acceptance of the manuscript:

1. As stated by authors, the detection of waterbodies was still conducted manually using drone images, with only minimal support from image recognition AI. Then I did not find any novelty in the manuscript. Until we are not detecting the breeding sites with automated AI enabled drone technology, it is not of much use. Further, drone images did not identify the sites positive for larvae. Even limited success in the same might have resulted in a lot of output.

2. The AI model section lacks methodological details, particularly regarding model validation and feature importance. How the CatBoost model was trained and validated (e.g., dataset split, hyperparameter tuning). Also provide limitations of the model’s precision and implications for potential overtreatment.

3. Use of 100 m height for drone application for time efficiency is not justified. As better resolution of images at 50 m height might have provided good results in larval site identification. Please clarify in detail.

4. There are no details of larvicide used in the manuscript. Kindly provide the same.

5. How the larval risk criteria were defined? Kindly provide.

6. Kindly provide details of how many habitats came under high risk larval sites. It will justify the claims of reduction in larvicide usage. Further, show the difference in area of breeding habitat sprayed both under conventional and targeted larviciding.

7. Please state why no ethical permission was obtained for drone based surveillance of breeding habitats. Whether the grid used in drone mappring (300x300m) area did not involve any human or cattle habitation. Further, state about ethical clearance for stakeholder and field worker Qualitative insights.

8. Why traps were placed at one site only in each conventional and interventional subdistricts. Traps placement in all the study location might provide better assessment of entomological outputs.

9. No statistical tests are presented to compare outcomes between groups, limiting confidence in the reported differences. The study would benefit from formal statistical testingto compare larvicide use, person-days, and mosquito abundance between groups. Present p-values or confidence intervals where appropriate to support claims of efficiency.

10. Line 459-484 seems not the part of results. It may be moved to discussion.

11. There is some inconsistency in results. Kindly present in clear and concise way. Specially line 533-594.

12. In discussion authors suggest mapping by drone and spraying manually. Both mapping and spraying by AI enabled drone technology could result in better output.

13. In methodology, there is no details of Feedback from field staff.

14. Kindly provide quantitative results for Qualitative insights from Local Stakeholders. How many field and stakeholder feedbacks were obtained. How many of those reported what.

15. In introduction, details of malaria cases of 2023 may be provided.

16. Legends for figure may be defined in better way.

**Reviewer #2: ** In recent years LSM has gained much attention since this method seems to be more promising. AI-based drone application is one such innovation in vector control. Some suggestions:

1. Here the larvicide was Vectolex WDG which is Bacillius spharicus 2362 (strain ABTS - 1743) was used. I do not fine its efficacy. Because its efficacy depends on exposure to sunlight, pH of water and in fragile ecosystem. Please give details.

2. Economic validation.

3. Community acceptance.

4. Impact on other insects.

5. Larval data missing.

7. Most mosquitoes were Culex. Impact of any diseases transmitted by Culex spp.

8. Limitations such as inability of breeding detection during dry season and also breeding habitats covered under canopy.

9. Whether other vector control agents such as larvivorous fish can be used?

**Do you want your identity to be public for this peer review?** For information about this choice, including consent withdrawal, please see our Privacy Policy

Reviewer #1: No

Reviewer #2: **Yes: ** Prof Susanta Kumar Ghosh

---

## [Author Response · Author response to Decision Letter 1]

19 Sep 2025

Manuscript ID: PONE-D-25-27483

Title: Field evaluation of drone and AI assisted larval source management in Ghana

・General Response

Dear Editor and Reviewers,

We sincerely thank you for your constructive comments and valuable suggestions on our manuscript. We have carefully revised the manuscript in response to all the points raised. Below, we provide a detailed point-by-point response. Reviewer comments are presented in **bold italics**, followed by our responses. All changes in the revised manuscript are indicated using track changes.

・Response to Journal Requirements:

1 When submitting your revision, we need you to address these additional requirements. Please ensure that your manuscript meets PLOS ONE's style requirements, including those for file naming.

Response:

We have revised the manuscript and files to comply with PLOS ONE’s style and formatting requirements, including the file naming conventions.

2 Please include a complete copy of PLOS’ questionnaire on inclusivity in global research in your revised manuscript.

Response:

As instructed, we have completed the questionnaire on inclusivity in global research and included it as Supporting Information in the revised submission.

3 Please note that PLOS ONE has specific guidelines on code sharing for submissions in which author-generated code underpins the findings in the manuscript. In these cases, we expect all author-generated code to be made available without restrictions upon publication of the work. Please review our guidelines at https://journals.plos.org/plosone/s/materials-and-software-sharing#loc-sharing-code and ensure that your code is shared in a way that follows best practice and facilitates reproducibility and reuse.

Response:

We acknowledge PLOS ONE’s policy regarding code sharing. In our revised manuscript, we have provided a detailed description of the AI methodology at a level sufficient to allow reproducibility of the study’s findings. However, we respectfully note that the underlying software implementation, including the trained AI models and author-generated scripts, forms part of proprietary assets developed by our company. To protect commercial interests, we are unable to make the raw code publicly available.

We believe that the methodological detail provided in the manuscript ensures transparency and scientific reproducibility, while the restriction on code release is necessary to safeguard intellectual property. We kindly request the editor’s consideration of this exemption in line with PLOS ONE’s data and materials policy.

4 Thank you for stating in your Funding Statement:

This study was supported by funding from the Japan International Cooperation Agency (JICA) and the Japan Science and Technology Agency (JST) under the framework of international technical cooperation. The funding was administered through SORA Technology Ltd., which coordinated the implementation activities. The funders had no direct role in the decision to submit the manuscript for publication.

Response:

As instructed, we have amended the Funding Statement to declare all sources of support and have included the required statement regarding additional external funding. The revised Funding Statement has been included in the cover letter.

5 Thank you for stating the following in the Competing Interests section:

This study was conducted with funding administered through SORA Technology Ltd., to which authors Masaki Umeda and Juhoe Kim are affiliated. Authors Godfred A. Bokpin, and Francis A. Adzei received research support through this funding mechanism. Authors affiliated with SORA Technology were involved in the study design, implementation, and manuscript preparation. All other authors declare no competing interests.

We note that one or more of the authors are employed by a commercial company.

Response:

As instructed, we have revised both the Funding Statement and the Competing Interests Statement to clearly declare the commercial affiliation with SORA Technology and to follow the required format. The amended versions have been included in the cover letter.

6 We note that your Data Availability Statement is currently as follows: All relevant data are within the manuscript and its Supporting Information files.

Response:

We acknowledge that the data underlying Figures 4, 5, 7, 8, 9, and 10 are required to allow replication of our results. Accordingly, we have prepared the relevant datasets and submitted them as CSV files under Supporting Information through the PLOS ONE submission system.

7 When completing the data availability statement of the submission form, you indicated that you will make your data available on acceptance. We strongly recommend all authors decide on a data sharing plan before acceptance, as the process can be lengthy and hold up publication timelines. Please note that, though access restrictions are acceptable now, your entire data will need to be made freely accessible if your manuscript is accepted for publication. This policy applies to all data except where public deposition would breach compliance with the protocol approved by your research ethics board. If you are unable to adhere to our open data policy, please kindly revise your statement to explain your reasoning and we will seek the editor's input on an exemption. Please be assured that, once you have provided your new statement, the assessment of your exemption will not hold up the peer review process.

Response:

We acknowledge the importance of ensuring timely data availability. In line with PLOS ONE’s open data policy, we have already prepared and submitted the minimal dataset required to replicate our findings. Specifically, the data underlying Figures 4, 5, 7, 8, 9, and 10 have been provided as CSV files and uploaded as Supporting Information through the submission system. These files will therefore be freely accessible upon publication.

8 Please include your full ethics statement in the ‘Methods’ section of your manuscript file. In your statement, please include the full name of the IRB or ethics committee who approved or waived your study, as well as whether or not you obtained informed written or verbal consent. If consent was waived for your study, please include this information in your statement as well.

Response:

As instructed, we have included a full Ethics Statement in the Methods section of the revised manuscript. The statement now specifies the approving institutions (the National Malaria Elimination Program and the Ministry of Health of Ghana), the process of obtaining informed consent from community representatives, and confirmation that no deviations from the approved study protocol occurred.

9 We note that Figures 2 and 3 in your submission contain map/satellite images which may be copyrighted. All PLOS content is published under the Creative Commons Attribution License (CC BY 4.0), which means that the manuscript, images, and Supporting Information files will be freely available online, and any third party is permitted to access, download, copy, distribute, and use these materials in any way, even commercially, with proper attribution. For these reasons, we cannot publish previously copyrighted maps or satellite images created using proprietary data, such as Google software (Google Maps, Street View, and Earth). For more information, see our copyright guidelines: http://journals.plos.org/plosone/s/licenses-and-copyright.

We require you to either present written permission from the copyright holder to publish these figures specifically under the CC BY 4.0 license, or remove the figures from your submission:

a. You may seek permission from the original copyright holder of Figures 2 and 3 to publish the content specifically under the CC BY 4.0 license.

Response:

We appreciate the editor’s concern regarding the use of copyrighted map/satellite images. In response, we have replaced Figures 2 and 3 with maps generated from OpenStreetMap, which is distributed under the Open Database License (ODbL) with map tiles licensed under CC BY-SA 2.0. We have revised the figure captions to include appropriate attribution (“Background map tiles © OpenStreetMap contributors, licensed under CC BY-SA 2.0. Additional annotations by the authors.”). No proprietary map sources (e.g., Google Maps or Google Earth) were used in the revised figures.

10 We note that Figure 1 in your submission contain copyrighted images. All PLOS content is published under the Creative Commons Attribution License (CC BY 4.0), which means that the manuscript, images, and Supporting Information files will be freely available online, and any third party is permitted to access, download, copy, distribute, and use these materials in any way, even commercially, with proper attribution. For more information, see our copyright guidelines: http://journals.plos.org/plosone/s/licenses-and-copyright.

We require you to either present written permission from the copyright holder to publish these figures specifically under the CC BY 4.0 license, or remove the figures from your submission:

“I request permission for the open-access journal PLO

---

## [Editor Report · Decision Letter 1]

24 Nov 2025

Field evaluation of drone and AI assisted larval source management in Ghana

PLOS ONE

Dear Dr. Kim,

Thank you for submitting your manuscript to PLOS ONE. After careful consideration, we feel that it has merit but does not fully meet PLOS ONE’s publication criteria as it currently stands. Therefore, we invite you to submit a revised version of the manuscript that addresses the points raised during the review process.

https://journals.plos.org/plosone/s/submission-guidelines#loc-laboratory-protocols . Additionally, PLOS ONE offers an option for publishing peer-reviewed Lab Protocol articles, which describe protocols hosted on protocols.io. Read more information on sharing protocols at https://plos.org/protocols?utm_medium=editorial-email&utm_source=authorletters&utm_campaign=protocols .

We look forward to receiving your revised manuscript.

Kind regards,

Johanna Pruller

Senior Editor

PLOS One 

on behalf of

Himmat Singh

Academic Editor

PLOS One

Journal Requirements:

Additional Editor Comments:

Although the authors have tried to answer most of the comments by reviewer 1, a few things regarding LSM should be further expanded upon, such as economic value and community acceptance as that adds dimensions to vector control.

---

## [Author Response · Author response to Decision Letter 2]

10 Dec 2025

・Response to Additional Editor’s Requirements:

Additional Editor Comments:

Although the authors have tried to answer most of the comments by reviewer 1, a few things regarding LSM should be further expanded upon, such as economic value and community acceptance as that adds dimensions to vector control.

Response:

We sincerely thank the Academic Editor for the constructive additional comments. In response to the request to further expand on aspects of larval source management (LSM), particularly the economic value and community acceptance, we have revised the manuscript to strengthen these dimensions and provide a fuller contextual understanding of the proposed drone- and AI-assisted workflow.

1. Economic value

To address the Editor’s comment, we have added a detailed discussion on the cost-related implications of implementing drone- and AI-assisted LSM. Specifically, in the Results section, we clarified the potential cost-effectiveness of the proposed approach. The newly added text explains that the workflow can reduce labor requirements by approximately half, aligning with prior evidence indicating that labor constitutes around 58% of LSM operational costs. We also describe how these reductions can improve scalability under fixed budgets, how initial drone investments may be recovered within roughly one year, and how improvements in spraying efficiency may decrease larvicide consumption. The added paragraph in discussion is as follows:

From an economic perspective, the findings of this study indicate that the introduction of drones and AI can potentially reduce labor costs by approximately half, even when implementing LSM at scales comparable to conventional methods. Previous study on LSM cost structures have reported that labor accounts for about 58 percent of total costs [43], suggesting that the proposed approach could substantially alleviate this major cost component. The magnitude of labor cost savings is expected to increase as the operational area expands, which implies that larger LSM coverage may be achievable within the same budget. In addition, for operations of a scale like that examined here, the initial investment required for drone procurement could be recovered within roughly one year through labor cost savings alone, indicating a favorable cost-effectiveness profile over the long term. Larvicide constitutes another considerable portion of LSM costs, accounting for around 14 percent [43]. Improvements in spraying efficiency can therefore reduce larvicide consumption, leading to lower operational costs per cycle. Overall, the proposed method may enable LSM implementation with fewer personnel and reduced larvicide use while maintaining or lowering total budget requirements, making it a promising strategy for sustainable vector control in resource-constrained settings.

2. Community acceptance

We also expanded the Discussion to include a fuller description of community engagement, resident perceptions, and operational safety considerations. We highlight the steps taken to ensure transparency and local cooperation, resident concerns regarding noise, crash risk, and privacy, and mitigation strategies implemented during field operations. We also contextualize drone noise levels relative to WHO guidelines and clarify how privacy was protected under the national regulatory framework. The added paragraph in discussion is as follows:

To support community understanding and cooperation, community engagement activities were conducted in each sub-district prior to implementation. Explanatory meetings with community elders were held to secure approval for the purpose, procedures, and flight plan of the LSM operation, and additional announcements were disseminated through local public speaker systems. Information sharing continued throughout the intervention period with support from the district NMEP office. Although some residents expressed hesitation or unease, no conflicts or operational disturbances were reported, and LSM activities proceeded smoothly, likely reflecting both prior community experience with LSM interventions and the effectiveness of early communication efforts.

Resident concerns regarding drone use centered on noise, crash risk, and privacy. With respect to noise, the sound of multirotor propellers may be perceived as unpleasant. However, the Mavic 3M was operated at approximately 100 m altitude, which substantially reduced sound pressure at ground level. Previous studies report that drones of similar size generate approximately 40 to 50 dB directly beneath the aircraft at about 120 m altitude [43]. These values fall within or below the World Health Organization daytime guideline thresholds for transportation noise, which are 53 dB for road traffic and 45 dB for aircraft [44]. Consistent with these expectations, no noise-related complaints were recorded. Even so, thresholds for noise annoyance vary across individuals and cultural contexts, indicating the need for future work that incorporates acoustic measurements and resident surveys. Crash risk cannot be eliminated entirely, but several mitigation measures were implemented. These included mandatory helmet use for field teams, restricting flights to an area of 300 m by 300 m that could be visually monitored, operating only with fully charged batteries, and avoiding adverse weather conditions. Regarding privacy, flight permission was obtained under regulations that prohibit secondary use of collected imagery. At 100 m altitude, the ground sampling distance was approximately 3 cm per pixel, which allows recognition of human presence but makes personal identification highly unlikely. Although household structures were visible, which introduces a residual risk of inferring household characteristics, such information can support malaria control activities such as planning for indoor residual spraying. Maintaining an appropriate balance between public health utility and privacy protection remains essential.

We believe these additions substantially enhance the manuscript by addressing the broader operational and social dimensions of LSM implementation, as requested. We sincerely appreciate the Editor’s guidance and hope that the revisions now meet the journal’s requirements.

---

## [Editor Report · Decision Letter 2]

26 Dec 2025

Field evaluation of drone and AI assisted larval source management in Ghana

PONE-D-25-27483R2

Dear Dr. Kim,

We’re pleased to inform you that your manuscript has been judged scientifically suitable for publication and will be formally accepted for publication once it meets all outstanding technical requirements.

Kind regards,

Himmat Singh

Academic Editor

PLOS One
---

## [Editor Report · Acceptance letter]

PONE-D-25-27483R2

PLOS One

Dear Dr. Kim,

I'm pleased to inform you that your manuscript has been deemed suitable for publication in PLOS One. Congratulations! Your manuscript is now being handed over to our production team.

Kind regards,

on behalf of

Dr. Himmat Singh

Academic Editor

PLOS One